# Projection of Expression Profiles to Transcription Factor Activity Space Provides Added Information

**DOI:** 10.3390/genes13101819

**Published:** 2022-10-08

**Authors:** Rut Bornshten, Michael Danilenko, Eitan Rubin

**Affiliations:** 1Department of Immunology, Microbiology & Genetics, Faculty of Health Science, Ben-Gurion University of the Negev, Beer Sheva 84105, Israel; 2Department of Clinical Biochemistry and Pharmacology, Faculty of Health Sciences, Ben-Gurion University of the Negev, Beer Sheva 84105, Israel

**Keywords:** RNA-seq data, reconstruction of transcriptional regulatory networks, RTN, transcription factors, acute myeloid leukemia, AML

## Abstract

Acute myeloid leukemia (AML) is an aggressive type of leukemia, characterized by the accumulation of highly proliferative blasts with a disrupted myeloid differentiation program. Current treatments are ineffective for most patients, partly due to the genetic heterogeneity of AML. This is driven by genetically distinct leukemia stem cells, resulting in relapse even after most of the tumor cells are destroyed. Thus, personalized treatment approaches addressing cellular heterogeneity are urgently required. Reconstruction of Transcriptional regulatory Networks (RTN) is a tool for inferring transcriptional activity in patients with various diseases. In this study, we applied this method to transcriptome profiles of AML patients to test if it provided additional information for the interpretation of transcriptome data. We showed that when RTN results were added to RNA-seq results, superior clusters were formed, which were more homogenous and allowed the better separation of patients with low and high survival rates. We concluded that the external knowledge used for RTN analysis improved the ability of unsupervised machine learning to find meaningful patterns in the data.

## 1. Introduction

Transcription factors (TFs) bind specific DNA sequences and activate or repress the expression of corresponding genes. The genes controlled by a TF form a regulon. TFs’ activity, and the activity of their associated regulons, is hard to assess as assays that measure regulon activity need to be optimized for each TF separately. The difficulty of assessing TF activity stems from the plethora of control mechanisms that determine this activity: the regulation of transcription level (e.g., FOXO1 [1]), phosphorylation (e.g., STAT3 [2]), the inhibition by binding proteins (e.g., KLF2 [3]), localization to the nucleus (e.g., [4]) or direct ligand binding in the nucleus (e.g., ESR [5]). As a result, there is currently no experimental method to assess the activity levels of many TFs in a single assay.

A computational method was described that can reconstruct regulon activity levels from global expression profiles. This method is based on a database that identifies TFs. It uses global expression profiles, either from RNA-seq or microarrays, to estimate the pattern of regulon activity that best explains the given expression pattern. This tool is readily available as an R package called “RTN: Reconstruction of Transcriptional regulatory Networks and analysis of regulons” (RTN) [6,7].

Leukemia is a group of blood cancers that develop in the bone marrow and result in high numbers of abnormal white blood cells (leukemic blasts) [8]. In this study, we examined expression data from acute myeloid leukemia (AML) patients. AML is the most aggressive type of leukemia and is characterized by the accumulation of highly proliferative blasts with a disrupted myeloid differentiation program. Several subtypes of AML have been defined clinically using The French–American–British (FAB) classification method, which is based on the type of cell from which the leukemia is developed and the level of maturity of the cells [9]. Other classification schemes have been proposed, e.g., the one suggested by the World Health Organization (WHO), which considers chromosome translocations and evidence of dysplasia [10]. According to the FAB classification, patients with PML/RARA translocation are classified as M3 AML patients. However, according to the most current WHO classification, these patients are classified as Acute Promyelocytic Leukemia (APL) and not as AML.

In AML, 60–80% of younger patients achieve complete remission after standard cytotoxic chemotherapy with high-dose cytarabine (cytosine β-D-arabinofuranoside; AraC) and daunorubicin. Unfortunately, most are expected to relapse and die, most likely due to the outgrowth of residual therapy-resistant AML cells in the bone marrow [11]. In older patients, which constitute the majority of AML patients, the situation is even worse due to the high toxicity, making a high dose of cytarabine unsuitable. Thus, low-dose Ara-C (LDAC) treatment is often given to unfit patients with unremarkable outcomes [12,13]. Even though several targeted AML drugs have been recently approved [14,15], their impact on long-term survival is yet to be determined. However, the approval of targeted therapies for specific subsets of AML patients marks the beginning of a personalized treatment approach that could yield better outcomes [14,16]. Currently, the average median survival time of patients with AML is about 8.5 months, and the 5-year overall survival rate is about 24% [17]. APL, on the other hand, is treated with all-trans retinoic acid (ATRA; tretinoin) and arsenic trioxide, and has a much better prognosis: the 5-year overall survival of APL patients is up to 90% [18].

Despite the remarkable genotypic and phenotypic heterogeneity of AML cells among patients, most AML studies investigate averaged responses at the population level, making the development of targeted therapy very challenging. It has been shown that different molecular subtypes of AML tumors, which can be pathologically indistinguishable, respond differently to the same drug regimens [19]. Thus, superior methods allowing matching patients to existing and investigational therapeutics are in high demand [20]. Deriving new features from expression profiles can provide new ways to divide AML patients, paving the way for the definition of more homogenous populations.

In this work, we directly assessed the contribution of RTN-derived features to the clustering of AML patients, using the older FAB classification. We considered three clustering methods and compared their performances using a quality of clustering measure (namely, Silhouette Coefficient) and a biological measure, namely, survival. Our results support the hypothesis that RTN-derived features contain information that cannot be teased from RNA-seq data with standard clustering methods. This finding is further supported by recreating these results in an independent dataset. In future studies we plan to turn to single cell analyses, where RTN-like approaches may help predict the responses of single cells to therapy. However, in this work, we exclusively use bulk RNA.

## 2. Methods

### 2.1. RNA-seq Data

Population 1: TCGA LAML. The TCGA-LAML [21] data were downloaded from cBioPortal (downloaded on 26 January 2020) or using the R package TCGAbiolinks that directly perform queries in the GDCquery (accessed on 29 November 2020). The TCGA-LAML includes mRNA expressions of 173 AML patients (of which survival data are available for 161 patients). The data downloaded from TCGAbiollinks included only 145 known patients (of which only 134 had survival information). A complete description of this population is provided elsewhere [21]. Briefly, the characteristics of the patients used in this study is given in Table 1 below.

Population 2: OHSU AML. The OHSU AML [22] data were downloaded from cBioPortal (downloaded on 13 December 2020). The OHSU AML includes the mRNA expressions of 451 AML patients (of which survival data were only available for 411). A complete description of this population is provided elsewhere [22]. Briefly, the characteristics of the patients used in this study are given in Table 1 below.

### 2.2. RTN

The RTN package (version 2.15.1) was run using R (R Core Team 2012). While a complete description of RTN can be found elsewhere [6,7], a brief explanation is provided for convenience. Briefly, the package tests the association between a given TF and all potential targets using transcriptome data from RNA-Seq or microarray. GDCquery was used to import data from TCGA with the following parameters: project = “TCGA-LAML”, data.category = “Transcriptome Profiling”, data.type = “Gene Expression Quantification”, workflow.type = “HTSeq–FPKM-UQ”.

### 2.3. Feature Selection

For population 1, feature selection was not used.

For population 2 the following procedure was used for feature selection: 75% of the data were randomly chosen for feature selection. Median survival was used to label patients as survivors or non-survivors (patients with the same survival time as the median were deemed “non-survivors”). The FeatureWiz package [23] of the Python programming language was used, with default parameters, to choose a subset of genes. Subsequent analysis was performed on the remaining 25%.

### 2.4. Clustering and Survival Calculations

Clustering algorithms were applied to the TCGA-AML mRNA expression dataset, including all the genes, with default parameters (k = 2 included). The resulting clusters were used to define patients’ sets, whose survival was compared using the Kaplan–Meier Survival plot and the log-rank test for the significance of difference in survival. To do this, we used the sklearn.cluster, kaplanmeier, matplotlib and fcmeans packages in the python programming language (version 3.8). Elbow analysis [24] suggested 3 groups, but we left k at 2 to facilitate further survival analyses.

To evaluate clusters’ quality, we arbitrarily chose the Silhouette Coefficient score. Other cluster quality measures such as the Rand Index, Adjusted Rand Index, Mutual Information, Calinski–Harabasz Index or the Davies–Bouldin Index were not tested.

## 3. Results

In a previous article, we showed that the RNA-seq dataset possess crucial information above and beyond the mutations’ data. In this work, we applied a related approach to test the hypothesis that RTN provides further information that cannot be learned from RNA-seq data. If true, this hypothesis predicts that supplementing or replacing expression profiles with RTN results should outperform RNA-seq data in machine learning tasks. To test this prediction, we chose to use unsupervised machine learning, comparing the grouping of AML patients in terms of survival with RNA-seq results alone or after adding RTN-derived regulon activities as features. For a detailed description of the study population see Table 1.

First, we identified the clustering method that is most suitable for the task at hand. For this reason, we compared the performances of three clustering algorithms (K-means [25], FCM (fuzzy C-means) [26] and BIRCH (Balanced Iterative Reducing and Clustering using Hierarchies) [27]) for unsupervised learning and evaluated their performance using two measures: the Silhouette Coefficient score and log-rank test for difference in survival. In the Silhouette Coefficient score test, the compactness of the resulting clusters is expressed as the average distance between patients in the same cluster and the average distance between points in different clusters (Figure 1). Our analysis suggests that the BIRCH algorithm gave slightly more compact clusters than the other two methods. Furthermore, when the survival of patients belonging to the clusters produced by the three methods was compared (Figure 2A), BIRCH was found to produce the best separation using RNA-seq features alone (*p*-value = 0.006, 0.0054 and 0.0014 in K-means, FCM and BIRCH, correspondingly). A visualization of the clusters can be seen in the Appendix A.

Next, we tested the clusters obtained with the BIRCH algorithms, with and without the results of regulon status inference by RTN (Figure 1 and Figure 2). This analysis indicated that clusters produced after adding regulon status (gray) identified groups of patients with more uniform survival times than clusters generated with RNA-seq data alone (*p*-value = 0.0014 compared to 0.008, log rank test, BIRCH algorithm). Interestingly, using only RTN-inferred regulon activity to cluster patients performed the worst in defining a patients group with distinct survival (*p*-value = 0.0038). When the separation into subtypes was compared between RNA alone and RNA with RTN (using the BIRCH algorithm to separate patients into groups), the groups were purer when RTN was added (Table 2). In fact, one of the groups contained only M3 patients, which are currently considered APL and not AML. These results suggest that adding the RTN features helped us rediscover the differences between APL and AML patients

These results were partially repeated in a second, independent validation cohort. Using a second dataset, we found that RTN helps define superior clusters, but only after applying feature selection using the FeatureWiz package in python and not for differences in survival (Appendix A). Finally, we tested an alternative hypothesis to explain our findings: it is not impossible that RTN simply adds dimension-reduced features to RNA-seq. This enhanced the ability of BIRCH to identify more biologically homogenous groups. If this is the case, any dimension reduction algorithm would lead to improved separation. However, our results do not support this alternative hypothesis (Figure 3): RNA-seq with PCA performed worse than RNA-seq with RTN features, both in terms of Silhouette Coefficient (Figure 3A; 0.3585 compared to 0.5151, respectively) and separation by survival (Figure 3B; 0.0014 and 0.0008, respectively). 

Taken together, our results show that AML patient grouping was most compact and biologically meaningful when RTN-derived features were added to RNA-seq data.

## 4. Discussion

In order to test if using RTN adds biologically meaningful information to RNA-seq profiles, we analyzed the data of patients with AML and compared the results of clustering in different ways based on RNA-seq data, with or without RTN-derived regulon data. Our approach was to compare the survival of patients assigned to different clusters using unsupervised learning of the different resulting datasets. We observed better results when RTN data were added to RNA-seq data than either dataset alone. From this analysis, we concluded that RTN provides information that is not contained in the gene expression data. Where does this information come from? Our results suggest that some of the added value of RTN is simply data-driven dimension reduction. Adding PCA engineered features to RNA-seq resulted in better-separated clusters, in terms of patient survival. Using RTN to derive features instead of PCA further improved the ability to create survival-distinct clusters (Figure 3). RTN utilizes results that are external to the experimental data under investigation (namely, the RNA-seq of AML cells). It uses the results of multiple ChIP-seq experiments to determine which gene is a TF and which is not. It also uses knowledge about the nature of genetic control to model and infer regulons. These results suggest that multi-omic approaches can be superior to techniques involving single omics, and that integrating prior knowledge could be beneficial in studying biological systems with machine learning.

To the best of our knowledge, this work represents the first attempt to cluster AML patients with a multiomic approach. Our results are not very strong: while we demonstrate that RTN adds information not found in RNA-seq data alone, the results are sensitive to feature selection and data choice (see below). This suggests that even though RTN adds novel information, RNA-seq with RTN is still not enough to classify AML patients into homogenous subtypes. Further research is required to determine what, if any, information should be added to RNA-seq data to achieve better subtype definition. It has not escaped our attention that in our analysis, RTN alone created more dispersed clusters than those created with RNA-seq alone in TCGA-LAML. This could be, at least in part, due to the shortcomings of the RTN approach. RTN uses data on TFs published previously in the literature [28]. However, some TFs are tissue-specific (e.g., GATA4 and TBX20, which are highly expressed only in cardiac muscle). So, taking all the TFs including those that are inactive in the cells under investigation can negatively impact our results. However, even if RTN could perfectly infer regulon activity, it is not impossible that other types of regulation are in play, such as alternative splicing, translational control, protein degradation and protein localization, which may impact tumor aggressiveness but may not be reflected in the regulon activity profile. RNA-seq may contain evidence of these processes that are not reflected in regulon activity, and as a result, using RNA-seq in addition to RTN-derived features may be beneficial for classifying cancer patients.

In the context of machine learning tasks, deriving regulon activity from expression profiles can be considered as knowledge-driven feature engineering. Prior knowledge of regulation networks, in the form of ChIP-seq results, is used to derive new features from RNA-seq results. The present work demonstrates that this is a viable approach to improving the performance of machine learning in biomedical tasks. While it is possible to learn from the data, the knowledge that can be derived from external sources might “consume” a large fraction of the data. Since in biomedical research data tend to be “fat” (i.e., have more features than observations), injecting external knowledge gained through carefully designed controlled experiments may be beneficial in cases where creating large datasets may be impossible due to limited access to samples (e.g., biopsies) or the costly procedures required to extend the datasets (e.g., RNA-seq). Thus, using RTN for feature engineering may be just a particular case of a more general collection of methods for prior knowledge-based feature engineering.

Can RTN aid machine learning to define purer subgroups simply by the dimensionality of this problem? Our results suggest that this is not the case. We showed that using a simple dimension reduction algorithm for feature engineering, namely, PCA, does not achieve the same improvement in the clustering of AML patients as RTN achieves.

Our results do not uncover a new subtype of AML patients. The cluster with a better prognosis after adding RTN features to RNA-seq is purely made of subtype M3 of AML patients, which is sufficiently distinct from other AML subtypes that it was later re-classified as APL. In fact, all the differences between patients assigned to this cluster and other clusters can be explained by the subtype/disease differences. However, we show that this distinction is more clear when RTN features are added.

This work focused on bulk RNA-seq, and we hope that when this study is performed on single-cell data, the outcomes will be more significant.

## 5. Conclusions

In this work we showed that RTN can be used to design new features that improve the performance of unsupervised machine learning methods in identifying biologically meaningful subtypes of AML patients. We showed that generating new features with RTN based on RNA-seq features improved the quality of clusters generated with both types of data (RNA-seq and RTN) compared to each type alone. We concluded that injecting prior knowledge into unsupervised learning, at least in the case of RTN and AML patient clustering, can help in identifying biologically meaningful subgroups.

## Figures and Tables

**Figure 1 genes-13-01819-f001:**
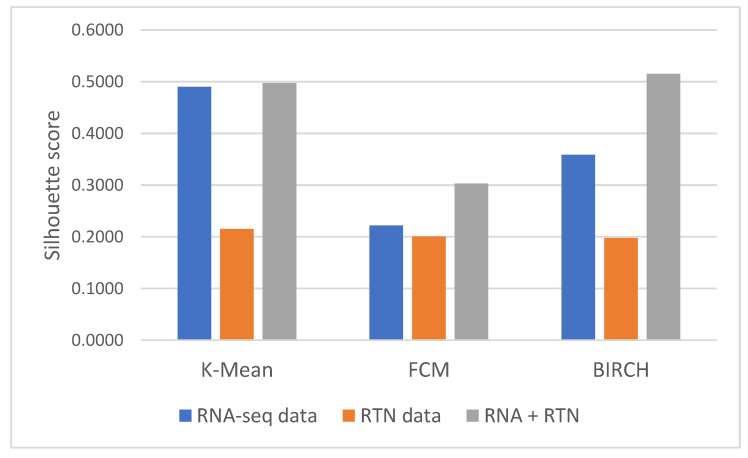
Silhouette Coefficient analysis of different features and clustering methods. Silhouette Coefficient scores were calculated for clusters generated with three methods (as denoted in the graph): K-means, FCM and BIRCH. Briefly, the Silhouette Coefficient reflects the ratio between the average geometric distance between patients in the different clusters to patients in the same cluster. Silhouette Coefficient values range from −1 to 1, with a value of 1 reflecting perfect separation between clusters. Each of the clustering methods was applied to three datasets, differing only in the features they involve: the first included only RNA-seq data (blue); the second included only RTN-derived features (Orange); and the third included both kinds of features (gray).

**Figure 2 genes-13-01819-f002:**
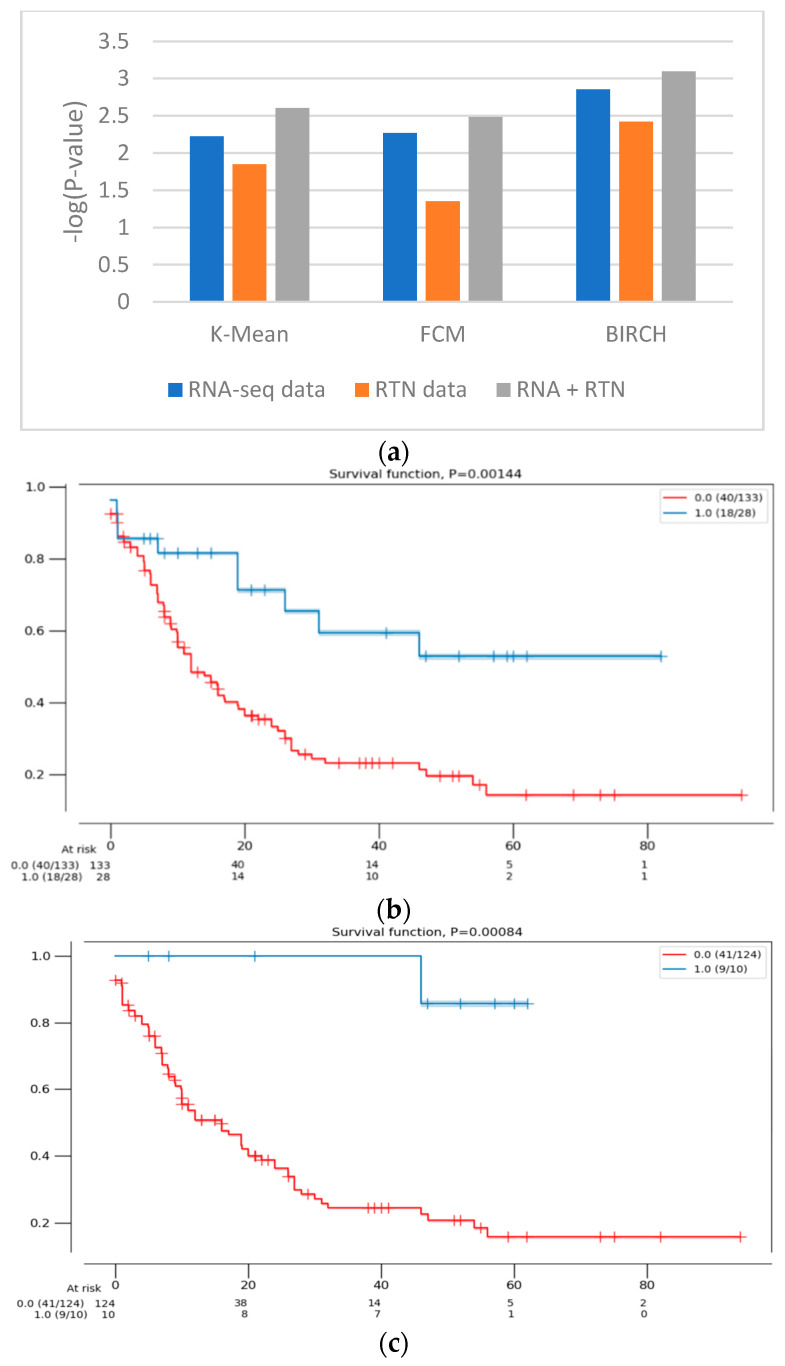
The difference in survival between the groups defined with different features. (**a**) The difference in survival was compared using the log-rank test for RNA-seq data (blue), RTN-derived features (orange) and both of them (gray). The clustering calculations were performed with 3 algorithms (denoted in the figure). The survival of the patients assigned to the two clusters was then compared, using the log-rank test. The difference in survival is expressed as -log(*p*-value) of the log-rank test, such that high values indicate a more significant difference in survival. (**b**) A Kaplan–Meier survival analysis plot of the clusters formed with RNA-seq data alone (161 patients), using the BIRCH clustering algorithm. Patients assigned to one cluster (red) are constructed with patients assigned to a second, smaller cluster (blue). (**c**) A Kaplan–Meier survival analysis plot of the clusters formed with RNA-seq- and RTN-based features (134 patients), using the BIRCH clustering algorithm. Patients assigned to one cluster (red) are constructed with patients assigned to a second, smaller cluster (blue).

**Figure 3 genes-13-01819-f003:**
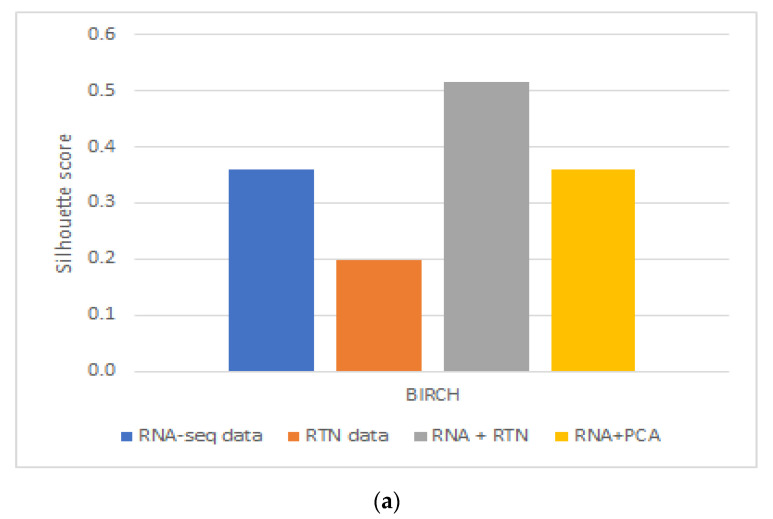
The impact of dimension reduction with PCA on cluster quality. The RNA-seq + RTN comparison data are compared to RNA with PCA. (**a**) Comparison of the Silhouette Coefficient. (**b**) Comparison of survival difference between patients assigned to each of the groups created with each data type.

**Table 1 genes-13-01819-t001:** Characteristics of the patients. The number of patients in each category is provided for both the discovery (TCGA) and independent validation set (OHSU).

Characteristic	TCGA	OHSU
Age at study entry—yr	55.3±16.1	57.0±18.1
N (with mRNA)		
Total	173	451
With survival data	161	411
With survival data and RTN	134	411
Race or ethnic group—no. (%)		
White	156 (90)	340 (83)
Black	13 (8)	14 (3)
Other	4 (2)	57 (14)
Male sex—no. (%)	93 (54)	233 (57)
AML FAB subtype—no.		
Undifferentiated AML—M0	16	6
AML with minimal maturation—M1	42	8
AML with maturation—M2	39	12
Acute promyelocytic leukemia (APL)—M3	16	10
Acute myelomonocytic leukemia—M4	35	26
Acute monocytic leukemia—M5	18	34
Acute erythroid leukemia—M6	2	-
Acute megakaryoblastic leukemia—M7	3	2
Not Classified	2	310

**Table 2 genes-13-01819-t002:** The FAB division of patients in the 2 groups. The BIRCH algorithm was used to divide the patients into 2 groups (see text), and the number of patients with each FAB type is shown. The likelihood of finding all patients in group 1 in M3 of the RNA + RTN analysis was calculated (*p* = 0.00371 for Newton’s binomial).

	RNA		RNA + RTN	
FAB	Group 0	Group 1	Group 0	Group 1
M0	16	-	15	-
M1	39	3	35	-
M2	31	8	36	-
M3	-	16	5	10
M4	34	1	28	-
M5	18	-	12	-
M6	2	-	2	-
M7	3	-	1	-
Not Classified	2	-	1	-

## Data Availability

The TCGA and OHSU AML datasets were downloaded from https://www.cbioportal.org/datasets accessed on 4 September 2022. The TCGA dataset was downloaded at 10 December 2021. The OHSU dataset was downloaded on 2 September 2020.

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
