# Peer review of "Projection of Expression Profiles to Transcription Factor Activity Space Provides Added Information"

_genes, 2022, doi:10.3390/genes13101819_

Round 1

Reviewer 1 Report

The Authors tested the usefulness of Transcriptional Regulatory Networks (RTN) data as a tool providing additional information for the interpretation of transcriptome data in patients with acute myeloid leukaemia.

Although these findings are interesting, several concerns need to be addressed before further consideration:

  - a more detailed characteristics of the studied patient group should be done (median age, number of patients over 60 years, sex, type of leukaemia – myeloblastic, promyelocytic, myelomonocytic, megakaryocytic, erythroblastic, type of the treatment applied – chemotherapy only, chemotherapy and stem cell transplantation). The aforementioned data will facilitate the interpretation of the results obtained due to the different molecular/cytogenetic characteristic of younger and older AML population group   - the final analysis was performed in terms of overall survival, but not in terms of leukaemia-free survival or leukaemia-related death. Therefore, the usefulness of the data obtained is limited and difficult to interpret    

Reviewer 2 Report

The manuscript describes an algorithm that combines the data obtained from RNA sequencing with the RTN-derived features to the better clustering of AML patients. This is an interesting and relatively new issue.
However, there are several points that I would like to comments:

1. The vast majority of cited researches refer to the Introduction. Have the Authors searched published articles for references presenting other proposed algorithms enriching the RNA-seq data of patients with leukemia or other cancers that they could compare with their approach to handling / processing biological data?

2. Figure 2 contains error in the caption. Point a) refers to the bar graph and the colors shown on it refer to the clustering algorithm, not to a data group. Point c) missing from Figure 2.

3. I cannot find in the Results section the information that was cited in the Discussion in lines 167-169. Perhaps it is worth adding tables with values for these parameters.

4. The presented difference in K-M survival plots between RNA-seq data alone and RNa-seq data with RTN is small (p values are similar). How the Authors explain the clinical usefulness of the proposed procedure? A more detailed description is needed in the Discussion section.

Reviewer 3 Report

The paper discusses use of Reconstruction of Transcriptional regulatory Network (RTN) in addition to RNA sequence data for clustering patient samples. Authors compare three clustering algorithms on different data types: RNA seq, RTN data and RNA seq with RTN detected regulons.

The authors used TCGA-LAML data that incudes mRNA expression of 173 AML patients and applied RTN to detect regulons in this dataset. These are then fed into the three machine learning algorithms and resulting clusters are compared using silhouette score and Kaplan-Meier survival plot. The results show regulons data (RTN) added to gene expression (RNA seq) has better scores, potentially adding prognostic information in AML.

Overall, I the paper is clearly written. More explanations are required in the methods to appreciate the robustness of the approaches. Also it should be discussed how clusters capture known genetic subgroups to understand how the work is additive to current understanding of AML.

 Major comments:

1-    Can authors explain the parameters used in each clustering algorithm, for example the number of clusters used in each, the threshold in Birch algorithm or fuzzy partition matrix exponent in FCM? Details should be added to methods.

2-    Did the authors perform robustness analyses using those parameters (e.g. numbers of clusters needed for optimal survival prediction). Have authors optimized the three algorithms by modifying parameters? I see k-mean of RNA seq dataset has quite a good silhouette score (fig. 1) and also all the algorithms on all the dataset (except FCM on only RTN data) show significance using log rank test. Can each algorithm be improved upon?

3-    A validation on an independent cohort other than TCGA would have been of important value (eg BEAT AML cohort PMID 30333627)

4-    In the survival plot, can authors explain the missing samples, out of 173 samples, we only see 161 samples in 2b and 134 in 2c, why is this? Are they not part of main clusters?

5-    Can authors explain samples biological significance of each clusters? Are these clusters enriched in some cytogenetic or mutational groups?

6-    The clinical importance of an integrated approach (RNA + RTN) is much debatable if less than 10% of samples can be identified as favourable (2c, only 10 samples). Were those favorable samples identified by other favourable features such as cytogenetic or mutation (based on ELN recommandations)? Are any adverse-risk patients reclassified in favourable using your approach?

7-    In the abstract line 14, authors discuss the urgent need of treatment approaches addressing cellular heterogeneity before introducing RTN. As RTN are used on bulkRNA, in what way are they addressing the question of cellular heterogeneity?

Minor comments:

1-    In figure 2, the labels a, b and c are assigned incorrectly.

2-    In discussion, line 167, the authors use the term knowledge-free dimension reduction, am not sure what do they mean by ‘knowledge-free’?

Reviewer 4 Report

You have clearly stated the purpose of your study. You have presented adequately your results to defend your proposal, that knowledge-driven machine learning improve the algorithm in biomedical tasks. The discussion is well organized and explains according to existing literature interesting observations derived from your analysis. I would like to highlight a few points mainly regarding the results section.

It is advisable to include as Suppl. Figures the visualization of clustering for each clustering method for each data set [RNA, RNA+RTN, RTN] and the Silhouette plot of each cluster plot together.

In legend of figure 1 and 2 you mentioned “Each of the clustering methods was applied to 3 datasets, differing only in the features they involve: the first included only RNA-seq data (blue); the second included only RTN-derived features (Orange); and the 3rd included both kinds of features (gray)”. However, in the figures the 3 colors refer to the three different clustering methods and no in the feature sets. Please, make the proper corrections.

In page 3 of 7, line 97 and 98, which is the article you referred to?

In page 3, line 112, you write that “BIRCH algorithm gave slightly more compact clusters than the other two methods”. Do you refer in general or for a specific dataset? In line 115 you mentioned “RNA-seq alone”. Is this applied for the previous phrase? And if so, based on the figure information, the figure 1 present lower Silhouette score for BIRCH algorithm and the highest in K-Mean clustering Method. The only case, where BIRCH has slightly higher SI and more compact, or well-defined clusters is for the RNA+RTN data set. On the other, based on the figure legend, where blue is RNA data and the first three bar refers to K-mean, the second to FCM and the third to BIRCH, then yes, the blue bar in the third group of bars is slightly higher… In any case I believe that the phrase in the lines 112 and 113 need clarification and the figures must be improved.

In page 3, line 114, please specify the figure you referred to, 2A.

Also I would like to ask you why do you prefer to use only Silhouette score and you didn’t calculate other commonly used indices (Rand Index, Adjusted Rand Index, Mutual Information, Calinski-Harabasz Index, Davies-Bouldin Index etc.) to compare the different clustering methodologies applied in the three data sets?

Round 2

Reviewer 1 Report

The paper was corrected and supplemented with additional data, as was suggested.

However, after the presentation of the more detailed characteristic of the study group, it was evident that the interpretation of the final data is likely incorrect. It is mainly due to the relatively high percentage of analyzed cases representing acute promyelocytic leukemia (AML M3), both in the discovery (TCGA) and independent validation (OHSU) sets.

It is important because AML M3 is associated with a good prognosis and an excellent response to the combined treatment with all trans retinoic acid and arsenic trioxide, which was not taken into consideration in the introduction and discussion section of the paper.

For these reasons, AML M3 should be analyzed separately or excluded from the study of the impact of the Reconstruction of Transcriptional regulatory Networks (RTN) data on survival rates.

Author Response

We thank the reviewer for taking the time to see the changes we made.

We did not claim that we define a new classification, but that adding the RTN helps to discover the existing classification.

Four sentences were added to the manuscript to refine this point. Sentences 1&2 were added to the introduction section, sentences 3 to the results section, and 4 to the discussion.

  1. According to the FAB classification, patients with PML/RARA translocation are classified as M3 AML patients. However, according to the most current WHO classification these patients are classified as Acute Promyelocytic Leukemia (APL) and not as AML.
  2. APL, on the other hand, is treated with all-trans retinoic acid (ATRA; tretinoin) and arsenic trioxide, and has a much better prognosis: 5-year overall survival of APL patients is up to 90% [18].

  3. In fact, one of the groups contained only M3 patients, which are currently considered APL and not AML. These results suggest that adding the RTN features helped rediscover the differences between APL and AML patients

  4. Our results do not discover a new subtype of AML patients. The cluster with a better prognosis after adding RTN features to RNA-seq is purely made of subtype M3 of AML patients, which is sufficiently distinct from other AML subtypes that it was later re-classified as APL. In fact, all the differences between patients assigned to this cluster and other clusters can be explained by the subtype/disease differences. However, we show that this distinction is made more clearly when RTN features are added.

Round 3

Reviewer 1 Report

To the Authors,

after the revision, the paper is far more clear and in agreement with the present state of knowledge on the subject.

I have no further comments and suggestions.